# On the phenomenon of the blue Sun

Nellie Wullenweber[1], Anna Lange[1], Alexei Rozanov[2], and Christian von Savigny[1]

[1]Institute of Physics, University of Greifswald, Felix-Hausdorff-Str. 6, 17489 Greifswald, Germany
[2]Institute of Environmental Physics, University of Bremen, Otto-Hahn-Allee 1, 28359 Bremen, Germany

**Correspondence:** C. von Savigny (csavigny@physik.uni-greifswald.de)

**Abstract.** This study examines the cause of the blue colour of the Sun as observed after the eruption of Krakatoa in 1883 as well as other volcanic eruptions or massive forest fire events. Aerosol particles, e.g., volcanic ash or products of biomass burning are believed to be able to modify the spectral distribution of transmitted solar radiation making it appear blue or green to a human observer.

Previous studies already showed that narrow aerosol particle size distributions with radii on the order of about $500\,\mathrm{nm}$ can lead to anomalous scattering, i.e., scattering cross sections increasing with increasing wavelength in the visible spectral range. In this work we treat the effect of Rayleigh scattering on the shape of the transmitted solar spectrum correctly employing radiative transfer (RT) simulations with the SCIATRAN RT-model. The colour associated with solar transmission spectra is determined based on the CIE colour matching functions and CIE chromaticity values. It is shown that a blue Sun can be

simulated for aerosol optical depths (at $550\,\mathrm{nm}$) of about $\tau = 0.5$ (or higher) if Rayleigh scattering is taken into account. Without considering Rayleigh scattering – as in most of the previous studies – a blue Sun is in principle produced with aerosol optical depths as low as about $\tau = 0.1$ (at $550\,\mathrm{nm}$), if the aerosol particle size distribution is chosen to maximize anomalous scattering in the visible spectral range. It is demonstrated that Rayleigh scattering – as expected – has a strong impact on the transmission spectrum, particularly at low solar elevation angles, and needs to be considered for a correct determination of

the perceived colour of the Sun. We also test the hypothesis that the blue Sun after the eruption of Krakatoa was caused by large abundances of water vapour in the atmosphere, as proposed in earlier studies. In addition, we present a case study on a particularly noteworthy blue-Sun-event in the past, i.e., the one related to the large Canadian forest fires in September 1950.

## 1 Introduction

This work has been inspired by various reports on sightings of the Sun appearing blue. Next to other reports, e.g., after the

extensive forest fires in Canada in 1950 (e.g., Gelbke, 1951; Wilson, 1951; Penndorf, 1953) widespread observations of a blue Sun have been made following the eruption of the Krakatoa volcano in 1883 (e.g., Larrabee, 1884; Kiessling, 1888; Symons et al., 1888). Witnesses described the Sun rising in a "splendid green", and turning "bright blue" near the zenith later during the day in the aftermath of the Krakatoa eruption as noted in Symons et al. (1888). Particularly noteworthy is the book by Kiessling (1888) that provides a comprehensive compilation of unusual atmospheric optical effects associated with the

Krakatoa eruption and also during the 2000 years before, including several reports on green and blue suns. Historical reports on an unusually coloured sun or moon can provide important independent information on exceptional natural events, e.g.,

volcanic eruptions (e.g., Bauch, 2017). Blue suns and moons are also discussed in standard textbooks on atmospheric optics (e.g., Bohren and Huffman (1998), section 4.4; Van de Hulst (1957), section 20.23).

Unfortunately, particle samples of unusual aerosol layers or reliable experimental evidence such as, e.g., spectral distribution measurements are typically not available for blue Sun events (exceptions are discussed below). For the Sun to appear blue, instead of white or yellowish, the normal spectral distribution of solar radiation has to be modified in a way, that less red light is transmitted to the observer compared to the blue component of light. The suppression of red light in transmitted spectra may in principle be caused by wavelength selective absorption or scattering processes, where absorption could be due to gaseous constituents or aerosols. The latter possibility is unlikely, because absorption coefficients of typical tropospheric aerosols usually decrease with increasing wavelength in the visible spectral range (e.g., Bergstrom et al., 2007). For this reason we focus on the possibility that the effect is caused by aerosol scattering. Previous studies have found that particles in the radius range from 400 – 700 nm (depending on the real part of the refractive index $n_r$) may lead to anomalous scattering, i.e., scattering coefficients increase with increasing wavelength in the visible spectral range, and could cause the observed effect if the aerosol particle size distribution is sufficiently narrow (Penndorf, 1953; Wilson, 1951; Porch, 1973; Ehlers et al., 2014). These studies, however, typically neglect the effect of Rayleigh scattering by air molecules on the spectral distribution of transmitted solar radiation. Rayleigh scattering suppresses the blue component, making it more difficult to produce a blue Sun.

It is also noteworthy that a blue Sun during a sunset on Mars has been captured in a picture taken by the Mars expedition rover Spirit in 2005, increasing the focus on the aerosol layer of the thin Martian atmosphere in order to find an explanation for the phenomenon (Ehlers et al., 2014).

Aerosol particles vary greatly in size but volcanic sulfate aerosol particles (e.g., von Savigny and Hoffmann, 2020) or biomass burning aerosols typically have sizes similar to the wavelength of visible radiation. If the aerosol particles are assumed to be spherical for simplicity, their scattering properties can be calculated according to Gustav Mie's scattering of light theory (Mie, 1908). In this study we assume spherical particles and Mie theory was used to model extinction coefficients of aerosol particles with different particle size distributions.

In this work, the full radiative transfer of solar radiation through the atmosphere was simulated with SCIATRAN, a radiative transfer model developed by the Institute of Environmental Physics of the University of Bremen, Germany (Rozanov et al., 2014). In addition, the colours resulting from the calculated transmission spectra are determined and displayed. The calculated spectral distribution reaching the Earth's surface has to be transformed into a colour on a digital output device, in a way a human with normal eyesight would perceive it. Colour modelling presented by the *International Commission on Illumination (CIE)* and the sRGB colour space was chosen and applied, the resulting colours are presented (CIE, 2004; Stokes et al., 1996).

The paper is structured as follows. Section 2 provides a short summary of the approach and methods used, describing both the radiative transfer model as well as the colour modelling. In section 3 the main results – including an investigation of the effects of Rayleigh scattering and absorption by $H_2O$ as well as a case study based on spectral observations after the intense Canadian wildfires in September 1950 – are presented. Sections 4 and 5 discuss and summarize the findings on possible characteristics of particles responsible for the blue colour of the Sun.

## 2 Methodology

We first describe the Mie simulations and the radiative transfer model employed, followed by a summary of the colour modelling approach used in this study.

### 2.1 Radiative transfer simulation: Mie scattering calculations and SCIATRAN

In order to model the transmitted solar radiation reaching the Earth's surface, first extinction coefficients of the aerosol particles were calculated using Mie theory and later implemented as an aerosol layer in the radiative transfer model SCIATRAN. In order to perform the Mie scattering calculations several assumptions have been made beforehand. We assume that the aerosol particle size distribution can be described by a mono-modal log-normal distribution:

$$n(r) = \frac{N_0}{\sqrt{2\pi} \cdot \ln(S) \cdot r} \cdot exp\left[-\frac{(\ln r - \ln r_m)^2}{2\ln^2(S)}\right] \tag{1}$$

Here, $N_0$ is the total particle number density, $r_m$ the median radius, $r$ the particle radius and $S$ represents the geometric standard deviation of the distribution, a measure of the distribution width (Grainger, 2017).

Mie simulations were carried out for various combinations of distribution width $S$, refractive index $n_r$ and median radius $r_m$, as described in more detail below. The Mie scattering calculations were carried out using routines from the *Earth Observation Data Group (EODG)* of the *University of Oxford*, available as downloads (for the software *IDL*) on their website (http://eodg. atm.ox.ac.uk/MIE/#intro, last checked: March 1, 2021). The choice of input values for the refractive indices and the median radii were guided by previous studies and literature (e.g., Ehlers et al., 2014; Kravitz et al., 2012; Penndorf, 1953). In this study we examined real parts of the refractive index $n_r$ between 1.3 and 1.5 and particle radii ranging from 25 to 3000 nm. The imaginary part of the refractive index was set to zero ($n_i = 0$) for most of the simulations presented in this study. We investigate absorption by aerosols as a potential cause of the blue sun phenomenon in section 3.9.

Having calculated the extinction coefficients of the aerosol particles possibly leading to a blue Sun, the impact of these aerosols and of air molecules on the transmission of solar radiation through the atmosphere was determined using the radiative transfer model SCIATRAN (Rozanov et al., 2014). SCIATRAN was developed by the Institute of Environmental Physics (IUP) of the University of Bremen for the interpretation of remote sensing measurements in the UV-visible-NIR-SWIR spectral range. In the transmission modelling mode, SCIATRAN can be used to simulate the atmospheric transmission of sunlight reaching the Earth's surface. In addition to the light scattering and absorption due to atmospheric gases, an aerosol layer can be included in the calculations. This layer is represented by the calculated extinction coefficient spectra covering a certain altitude range. The viewing geometry is set up with SCIATRAN in a way that the imaginary observer on the Earth surface is looking directly into the Sun. The solar zenith angle (SZA) as well as the optical depth have to be considered as possible influencing factors. For more information on how to use SCIATRAN see the detailed *User's Guide* provided as a download on the website of IUP Bremen (IUP, 2018). The model output contains transmission values at different wavelengths and solar zenith angles. Using the

SORCE solar irradiation spectrum (LASP, 2003) the resulting spectral distribution of solar radiation through an aerosol-loaded atmosphere can then be calculated.

## 2.2 Calculation and display of colour values

The colour corresponding to a given solar transmission spectrum was determined and displayed using a standard approach (e.g., CIE, 2004; Brainard and Stockman, 2010). Based on the CIE (International Commission on Illumination) $\overline{x}(\lambda)$, $\overline{y}(\lambda)$ and $\overline{z}(\lambda)$ colour matching functions, which quantify the spectral sensitivity of the cone cells of the human eye, we determined the so-called XYZ tristimulus values. The tristimulus values correspond to the spectrally integrated transmission spectrum weighted with the corresponding colour matching function (we refer to CIE (2004) for more detailed information). Based on

the XYZ tristimulus values the CIE chromaticity values $x$ and $y$ are calculated using

$$x = \frac{X}{X+Y+Z} \qquad y = \frac{Y}{X+Y+Z} \tag{2}$$

$x$ and $y$ are then displayed in a 2-D plot, as, e.g., in the right column of Fig. 3.

In addition we convert the tristimulus values to sRGB (standard RGB), which can be directly used to display colours in IDL (*Interactive Data Language*), the software used to prepare all Figures shown in this study. The conversion was done according

to the *International Color Consortium (ICC)* (see Stokes et al., 1996) with a linear matrix transformation and a non-linear gamma correction. The colours of the monochromatic colour arc – shown as the arc from blue over green to red in the right panels of Fig. 3, which will be discussed in detail below – as well as the colours corresponding to the transmitted solar spectra were determined this way. The procedure was verified using several external routines. Different websites offer colour value calculators and the display of colours according to their sRGB-values (e.g., http://www.brucelindbloom.com, last checked:

March 1, 2021; https://www.w3schools.com/colors/colors_rgb.asp, last checked: March 1, 2021).

## 3 Results

As mentioned above, in many previous studies on the origin of a blue moon or Sun, enhanced scattering of red light is provided as an explanation of the phenomenon. Usually, only scattering by the aerosol particles is considered, and Rayleigh scattering by air molecules is neglected. We start our considerations based on this assumption (Section 3.1) and then include a correct

treatment of Rayleigh scattering based on SCIATRAN radiative transfer simulations in section 3.2.

### 3.1 Results without Rayleigh scattering and gaseous absorption

Previous studies, e.g., Penndorf (1953) suggest that the particle population leading to the blueing effect has a relatively narrow particle size distribution and a refractive index (real part $n_r$) of 1.3 to 1.5. Figure 1 shows the Mie extinction coefficient as a function of wavelength and median radius (normalized to the value at a reference wavelength of 400 nm) for two different

distribution widths and for three different values of the real part of the refractive index. Results for a distribution width of $S$ = 1.05 (and $S$ = 1.2) are shown in the top (bottom) row. The left, middle and right columns of the Fig. display results for refractive indices of (1.3, 0·i), (1.4, 0·i) and (1.5, 0·i), respectively.

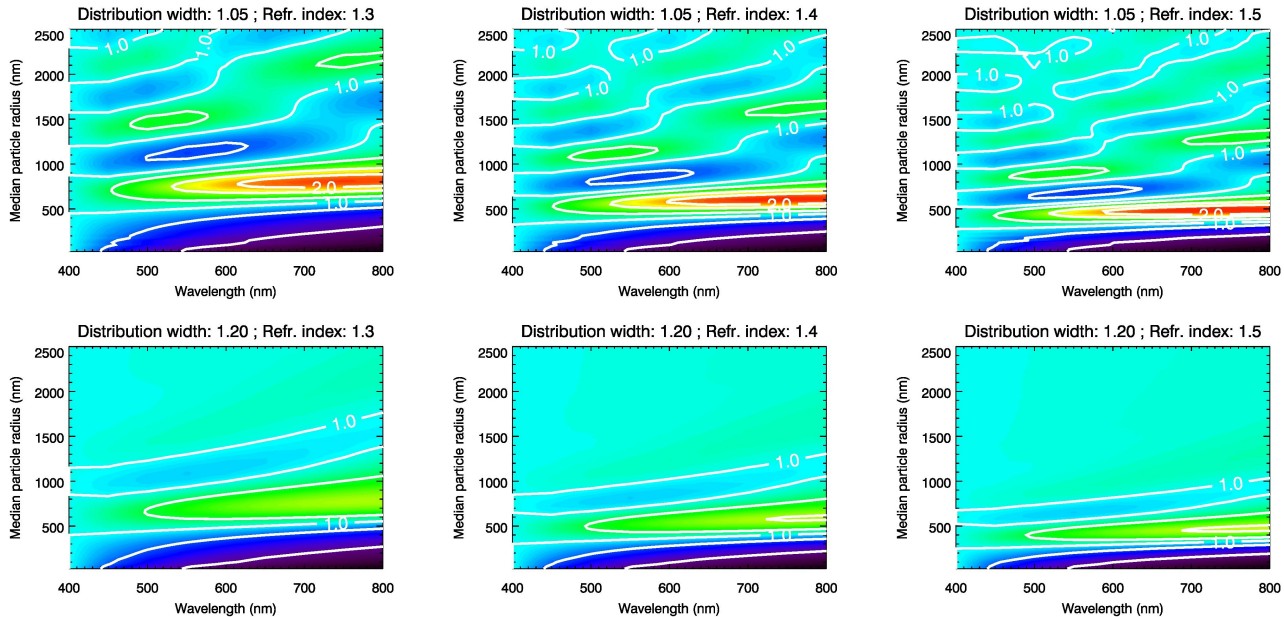

**Figure 1.** Spectral and radius dependence of aerosol extinction coefficients (normalized to 400 nm) determined by Mie-calculations assuming a mono-modal log-normal particle size distribution with a geometric width of S = 1.05 (top row) and S = 1.2 (bottom row) and real parts of the refractive index of 1.3 (left column), 1.4 (middle column) and 1.5 (right column). Contour levels are 0.3, 0.7, 1.0, 1.3, 1.7, 2.0, 2.3.

As can be seen in Fig. 1, the strongest increase in extinction coefficient with increasing wavelength in the visible spectral range (or maximum anomalous extinction) is obtained for median radii in the 400 – 700 nm range, with the specific values depending on the assumed refractive index. Two features of the plots are particularly noteworthy. First, the broader the size distribution, the less pronounced the anomalous extinction is (compare top and bottom row of Fig. 1). Secondly, the median radius producing the maximum anomalous extinction depends on the refractive index, decreasing from about 700 nm to about 450 nm as the real part of the refractive index increases from 1.3 to 1.5. In other words, depending on the refractive index, particles of a slightly different size will more easily produce a blue Sun. It is also noticeable that a lower refractive index leads to a larger particle radius range with stronger anomalous scattering, making the blue Sun phenomenon more probable. Unfortunately, no aerosol samples have been taken and examined from past events, therefore the true value of the refractive index of the particles producing a blue Sun is unknown. Penndorf (1953) lists in his study on sightings of the blue Sun in 1950 products of biomass burning with a refractive index of around 1.46. In his calculations however he uses the value of 1.33, based on the refractive indices of water and ice. Ball et al. (2015) have found real parts of refractive indices of volcanic ash of

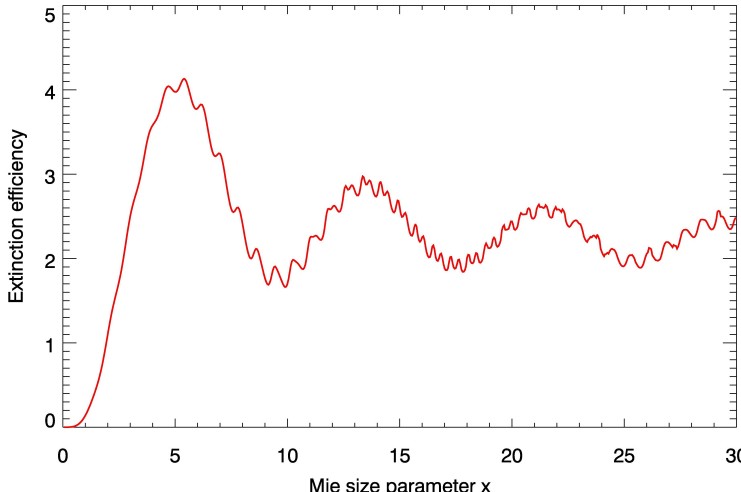

**Figure 2.** Dependence of the extinction efficiency on the Mie size parameter for monodisperse spherical particles with a refractive index (real part) of 1.4. For x-ranges with a negative slope, a blue sun may occur.

around 1.5. The results described in the following are based on a distribution width of $S = 1.05$ and a real part of the refractive index of 1.4, unless stated otherwise. These choices are arbitrary to a certain extent, but it is impossible to present results for all possible parameter combinations. In addition, the main conclusions of the study are not affected by the specific values of the distribution width and refractive index chosen.

In several publications (e.g., Horvath et al., 1994; Bohren and Huffman, 1998) the dependence of the Mie extinction efficiency on the size parameter $x = 2\pi r/\lambda$ is shown – rather than the spectral dependence of the extinction cross section as in Fig. 1 – to illustrate the parameter range for which a blue sun or moon may be expected. For completeness, we show such a diagram in Fig. 2. For x-ranges with positive slopes, the extinction efficiency decreases with increasing wavelength and vice versa, i.e. a negative slope may lead to a blue sun.

### 3.2 Rayleigh scattering and gaseous absorption

For the radiative transfer simulations carried out in this study, SCIATRAN version 3.8 (Rozanov et al., 2014) is used. Standard profiles of pressure, temperature and atmospheric trace gases (including $O_2$, $O_3$ and $H_2O$ that are relevant in the visible spectral range) for northern mid-latitudes taken from simulations with the Bremen 3-D CTM (Sinnhuber et al., 2003) are used, no clouds are included. Note that the choice of atmospheric background parameter profiles has a negligible effect on the results. SCIATRAN is run in *int - transmission - CDI* mode with refraction switched on, simulating directly transmitted radiation to an observer on the Earth's surface (IUP, 2018; Rozanov et al., 2014). The "int" mode is the mode allowing intensity or radiance calculations. The "transmission" mode enables the calculation of solar radiation transmitted through a spherical atmosphere and "CDI" means that the radiative transfer equation is solved with the combined differential-integral (CDI) approach (for more

details we refer to Rozanov et al. (2014)). The Rayleigh scattering cross section and its spectral dependence was taken from Bates (1984). For the purpose of comparison with later findings, the results of the radiative transfer calculations and colour

modelling of a Rayleigh-only atmosphere – i.e., without aerosols – are shown in Figure 3. This Figure is discussed now in more detail, because the following Figures are structured in the same way. The different rows of Fig. 3 show results for different SZAs ($30°$, $60°$, $80°$ and $90°$). The left column displays solar transmission spectra determined by multiplying the SORCE solar irradiance spectrum (LASP, 2003) by the transmission spectrum simulated with SCIATRAN. The plots in the right column of Fig. 3 show the chromaticity values $x$ and $y$ determined as described above. This is a standard way to depict colour information.

The arc with the filled colour circles represents the positions of the corresponding monochromatic spectra in the $x$-$y$-plane. The colours of the circles are based on the conversion of the chromaticity values to sRGB as described in section 2.2. The smaller open circle annotated by "$S_0$" corresponds to the chromaticity values of the unattenuated solar spectrum. The larger circle displays the position and the colour of the corresponding transmission spectrum in the $x$-$y$-plane. The colour of the Sun changes from whitish-yellowish for small SZAs to orange for SZA = $90°$, as expected. This is easily explained by the $\lambda^{-4}$

dependence of Rayleigh scattering (Strutt, 1871), i.e., radiation with shorter wavelength (blue light) is scattered more strongly than radiation with longer wavelengths. The pathlength of light through the atmosphere increases with increasing SZA, and the impact of Rayleigh scattering also increases, leading to an orange sunset which can be seen in the lowest panel of Fig. 3 where the Sun is at the horizon.

### 3.3 Simulation of a blue Sun

For the Sun to appear blue, the spectral irradiance values and therefore the transmission must be higher at shorter wavelengths relative to longer wavelengths in the visible spectral range. To reach this, in the following an aerosol layer was inserted into the model atmosphere and the radiative transfer calculations were repeated. Previous studies as well as the contour plot in Fig. 1 suggest that particles with a median radius of $400 - 700$ nm (depending on the refractive index) are able to cause a blue Sun. In this study different particle sizes in the range from 25 to 3000 nm have been tested and Rayleigh scattering taken into

account. Fig. 4 shows results in a similar way as Fig. 3, but with an aerosol layer in $6 - 8$ km altitude, an aerosol optical depth of $\tau = 1$, a real part of the refractive index of 1.4, median radius $r_m = 550$ nm and a distribution width of $S = 1.05$. Note that the layer altitude was found to have only a minor effect on the results, except for large solar zenith angles (see also section 4). According to Fig. 1 this set of parameters is associated with close to the maximum possible anomalous extinction in the visible spectral range. Fig. 4 shows that these assumptions lead to a blue colour of the Sun. Even for the smallest SZA of $30°$ a

light blue colouring of the Sun occurs. For the largest SZAs shown, the Sun assumes a deep blue colour. For increasing optical depth, the blueing effect is more pronounced for all SZAs, the transmission is further reduced and for an optical depth of, e.g., $\tau = 2$ the Sun assumes an indigo colour for SZAs of $80°$ and $90°$ (not shown). Another aspect that must be considered is that colours may not be perceived if the transmitted solar radiation is too weak. The irradiance values shown in the left column of Fig. 4 decrease significantly for increasing SZA. For a SZA of $80°$ the irradiance values are up to 2 orders of magnitude

lower than for SZA = $30°$ and decrease by several orders of magnitude for SZAs approaching $90°$. These values are of course only valid for the aerosol parameters assumed for the simulations presented in Fig. 4. According to Horvath et al. (1994),

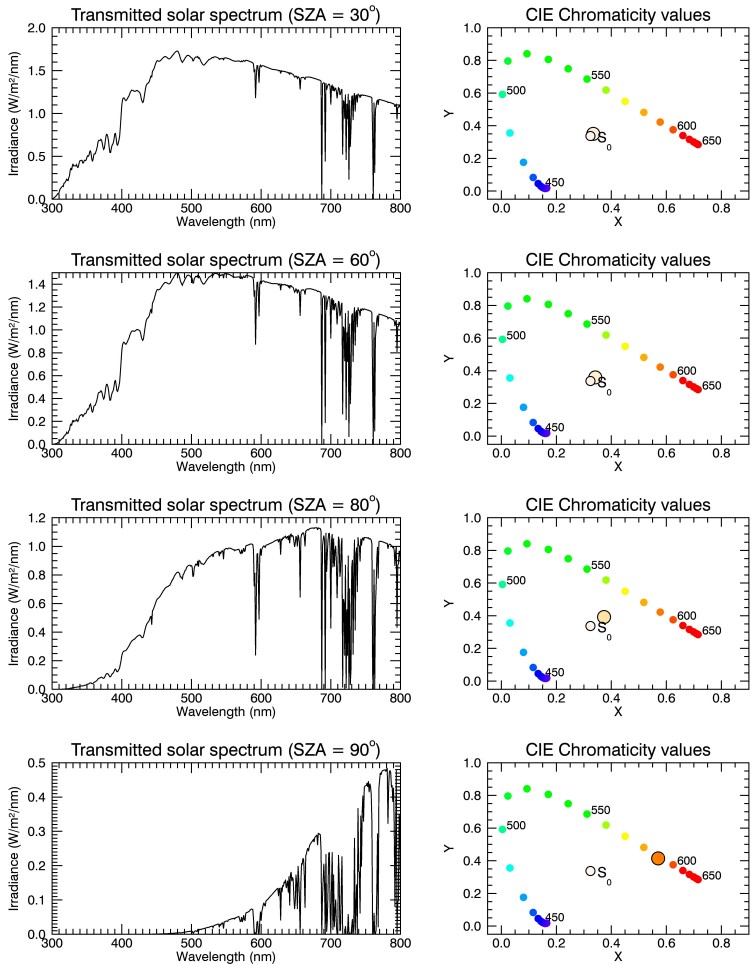

**Figure 3.** Results for an atmosphere without aerosols. The plots in the left column show the solar transmission spectra simulated for solar zenith angles of 30°, 60°, 80° and 90°, respectively (top to bottom), while the diagrams in the right column show the corresponding CIE chromaticity plots. The transmission spectra are calculated for an observer at the Earth's surface (0 m a.s.l.) looking directly into the Sun.

colour perception will not work if transmission values are smaller than about $10^{-7}$, which puts additional constraints on the combination of parameters producing a blue sun. In addition, colours may not be perceived if transmission is too large. Note that section 4 includes a discussion of the effect of contrast on colour perception.

### 3.4 Impact of particle size

Based on the results shown in Fig. 1, significant anomalous extinction in the optical spectral range can only be produced for a relatively narrow particle size distribution and for specific particle sizes, represented here by the median radius of a mono-modal log-normal distribution. Figure 5 shows chromaticity diagrams for different SZA and median radius values, for

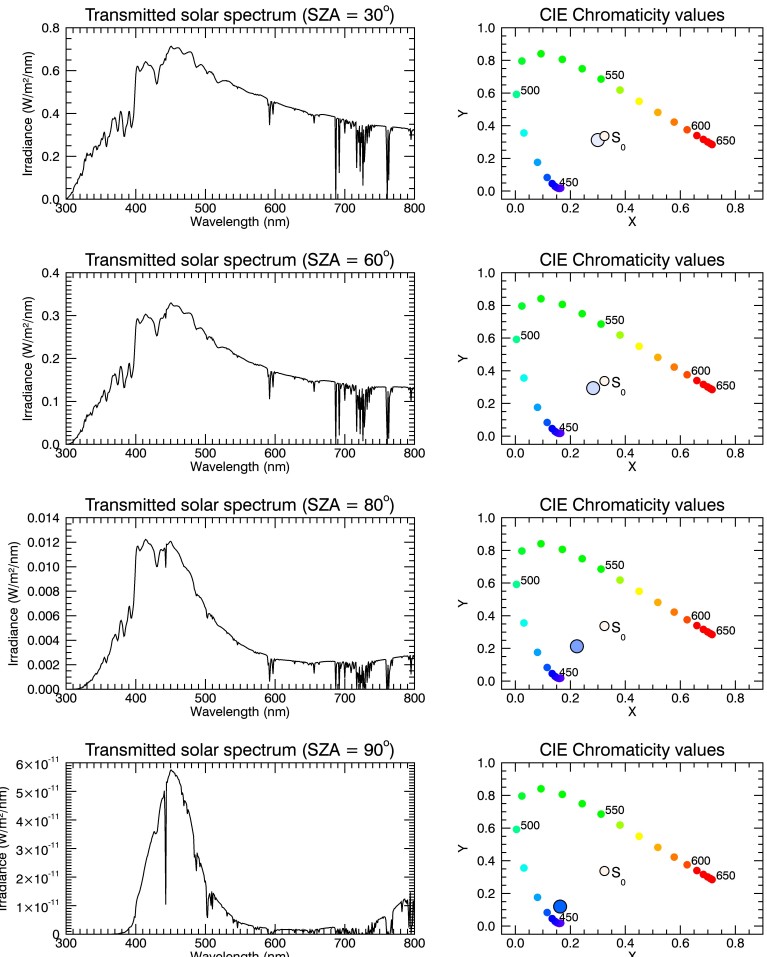

**Figure 4.** Simulated transmission spectra (left column) and CIE chromaticity plots (right column) for a ground-based observer looking directly into the Sun and for SZAs of $30°$, $60°$, $80°$ and $90°$ (from top to bottom). The simulations include Rayleigh scattering, molecular absorption by all relevant species and an additional aerosol layer with anomalous extinction and a vertical optical depth of $\tau = 1$ (at 550 nm), a distribution width of $S = 1.05$, a real part of the refractive index of 1.4 and a median radius of $r_m = 550$ nm. A blue Sun can be reproduced with the assumptions made for all SZAs considered.

an aerosol layer with an optical density of $\tau = 1$, a height of $6 - 8$ km and a refractive index of $n = 1.4 + i \cdot 0$. For a median radius of $r_m$ of 350 nm (first column of Fig. 5) the aerosol extinction coefficient decreases with increasing wavelength for the refractive index assumed here (see also Fig. 1) and the aerosol layer amplifies the effect of Rayleigh scattering and leads to a red Sun. Note that our results nicely confirm the general notion that a pure Rayleigh scattering atmosphere will only produce an orange sun during sunset (see Fig. 3), while aerosols with a sufficiently small size are required for red sunsets (left column of Fig. 5). The results for $r_m = 150$ nm (not shown) are quite similar to the results for $r_m = 350$ nm. The second column shows the

results for $r_m = 550$ nm, i.e., the median radius roughly associated with the most extreme anomalous extinction for a refractive index (real part) of 1.4, according to Fig. 1. A blue Sun can clearly be modelled with this set of parameters. Interesting are the results for a median radius of 750 nm (third column), which show a more and more green Sun with increasing SZA. For $r_m = 950$ nm (last column) the modelled Sun becomes again orange-red for the largest SZA, consistent with the spectral dependence displayed in Fig. 1.

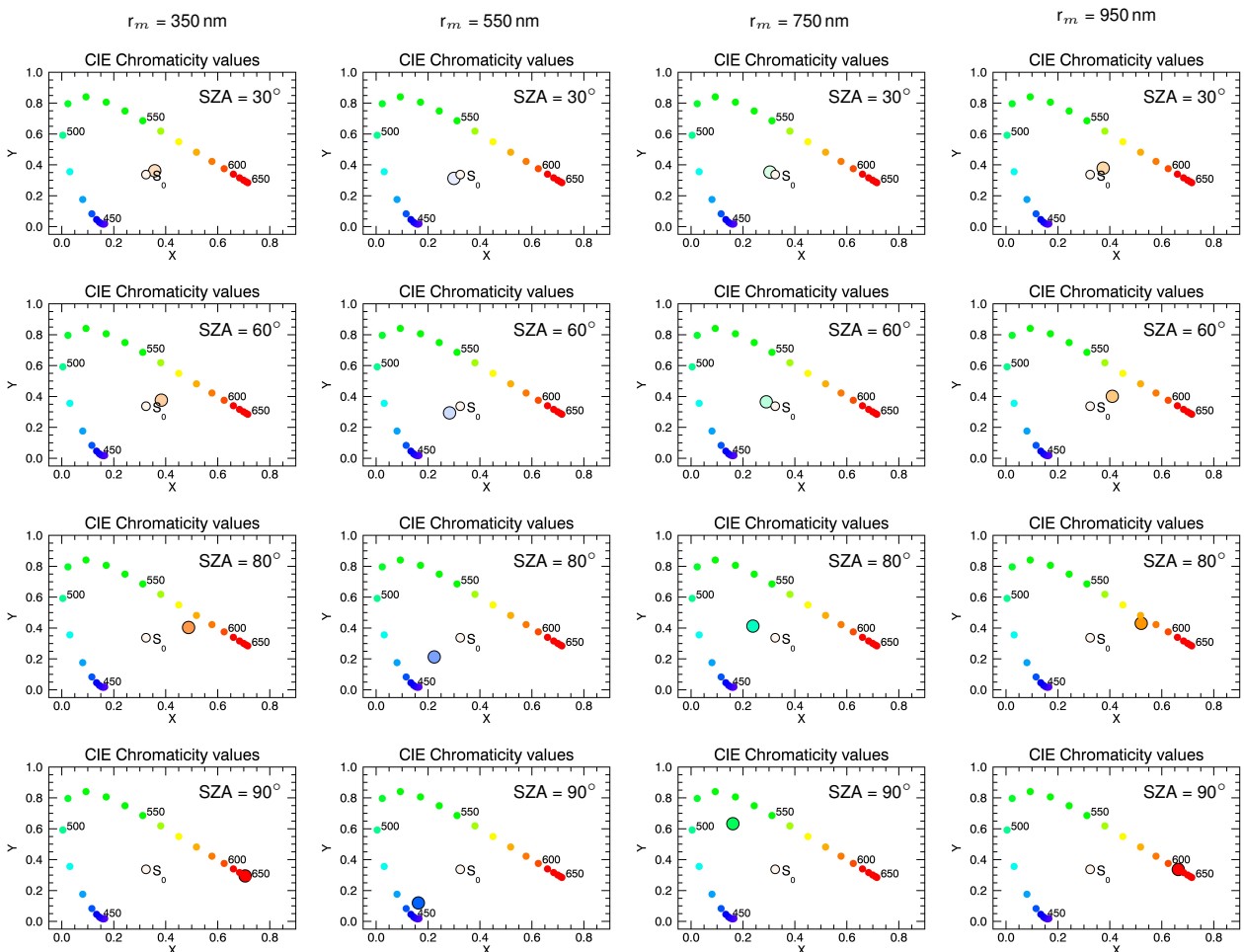

**Figure 5.** Chromaticity diagrams for different combinations of SZA and median radius $r_m$ of the mono-modal log-normal particle size distribution. Results for SZAs of 30°, 60°, 80° and 90° are shown in the different rows of the Figure, whereas the columns show results for $r_m$ of 350 nm, 550 nm, 750 nm and 950 nm. All results are for a distribution width of S = 1.05, a real part of the refractive index of 1.4, aerosol optical depth of $\tau = 1$ and an aerosol layer in the 6 – 8 km altitude range.

### 3.5 Aerosol scattering vs. Rayleigh scattering

As in most earlier studies on the phenomenon of the blue Sun Rayleigh scattering was not properly taken into account, it is worthwhile to investigate, how the results change, if Rayleigh scattering is switched off in the RT model. The results for an aerosol layer in $6 - 8$ km altitude with an optical depth of $\tau = 1$, median radius $r_m$ = 550 nm, distribution width $S$ = 1.05 and without Rayleigh scattering are shown in Fig. 6. Apparently and as expected, a blue Sun is more easily produced without Rayleigh scattering for all SZAs. Ignoring Rayleigh scattering, a blue Sun can be simulated for optical depths as low as approximately $\tau = 0.1$ for $r_m$ = 550 nm, $S$ = 1.05 and $n_r$ = 1.4 (results not shown). If Rayleigh scattering is considered, an aerosol optical depth of at least $\tau = 0.5$ is required to cause a blueing of the Sun. Note that depending on the specific set of aerosol parameters chosen, there are very large differences in CIE chromaticity values and colour, depending on whether Rayleigh scattering is considered or not. Unfortunately, only a fraction of the simulations carried out can be shown here.

One possibility to reduce the impact of Rayleigh scattering in the real atmosphere is to observe from higher altitudes, e.g., from an aircraft (e.g., Horvath et al., 1994). Under these conditions – and an aerosol layer occurring above the observer – Rayleigh scattering is significantly reduced, providing more favorable conditions for a blue Sun to occur.

### 3.6 Case study: Results by Wilson (1951)

Among the available studies on the 1950 Canadian forest fires (e.g., Schüepp, 1950; Ångström, 1951; Dietze, 1951; Gelbke, 1951; Jenne, 1951; Wilson, 1951; Penndorf, 1953), the work by Wilson (1951) is particularly noteworthy and relevant, because it contains a measured spectrum of transmitted visible radiation for a blue Sun scenario, obtained on September 27, 1950 from Edinburgh. The author also took measurements of a "normal" Sun spectrum at the same solar zenith angle, which allows the determination of the slant aerosol optical depth. Interestingly, the blue Sun and the normal Sun spectrum shown in Wilson (1951) (see Plate 8 on page 478) exhibit unexpected spectral features – maxima near 450 nm and 550 nm and a pronounced minimum around 490 nm – that differ significantly from the transmission spectra simulated in the present study (see also Fig. 8). We currently do not have a convincing explanation for these spectral features. The slant aerosol optical depth spectrum for a SZA of 71° based on the analysis of Wilson (1951) is shown in Fig. 7. It should be mentioned that Fig. 7 shows the optical depth in the sense of Beer-Lambert's law, i.e., $\tau = -\ln \frac{I}{I_0}$ and not the quantity shown in Fig. 1 of Wilson (1951). Note that spectral measurements were only available in the spectral range from 381 nm to 633 nm. Apparently, the optical depth increases from 450 to 650 nm, but the variation is with about 1 not very large. This spectral variation is smaller than in the simulations shown in Figs. 4 and 6. The optical depth spectrum of Wilson (1951) was read in by SCIATRAN and we performed radiative transfer simulations for the solar zenith angle (SZA = 71°) of the blue Sun measurement by Wilson (1951) in order to test, whether this optical depth spectrum does indeed produce a blue Sun. The aerosol layer was assumed to occur in $10 - 12$ km altitude, following Penndorf (1953). The vertical aerosol optical depth in this case was $\tau \approx 3.2$ at 550 nm based on the slant optical depth reported by Wilson (1951) and converted to vertical optical depth using SCIATRAN. The results are shown in Fig. 8 including Rayleigh scattering (top row) and without Rayleigh scattering (bottom row). A light blue colour of the Sun is already visible in the top row, i.e., including Rayleigh scattering, but the effect is more pronounced without Rayleigh scattering

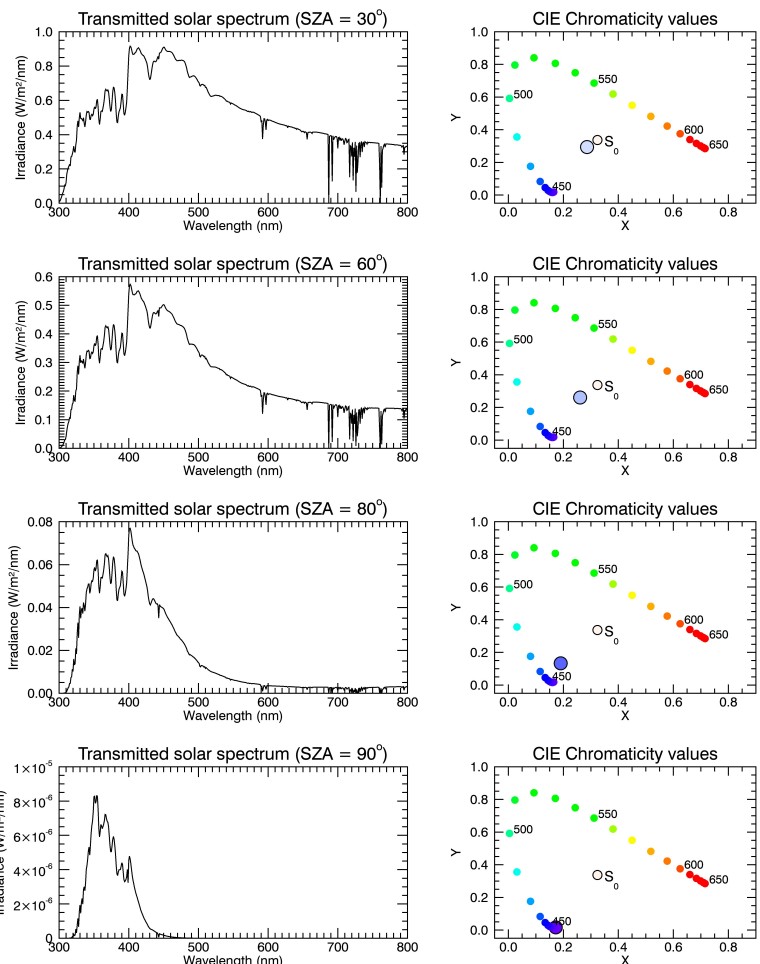

**Figure 6.** Similar to Fig. 4, but with Rayleigh scattering switched off. All other parameters are identical as in Fig. 4, i.e., $\tau = 1$, $r_m = 550$ nm, $S = 1.05$, layer altitude 6 – 8 km.

– as expected – again demonstrating that correct treatment of Rayleigh scattering is important. Neglecting Rayleigh scattering shifts the spectral maximum to slightly shorter wavelengths. With a slant optical depth of 9 – 10 (as in Fig. 7), solar radiation is attenuated by 4 – 5 orders of magnitude.

It is also noteworthy that Gelbke (1951) presented a photograph of the blue sun – taken from Greifswald, Germany by W. Gelbke on September 27, 1950, i.e., one day after the spectral measurement by Wilson (1951) – which, to our knowledge, is the only available photograph of the event.

The simulations show that the aerosol optical depth spectrum determined by Wilson (1951) does indeed allow simulating a blue Sun, although the minimum in the solar transmission spectra near 490 nm was not reproducible. A plausible explanation for these spectral signatures may be that both the "blue Sun" and the "normal Sun" spectra presented by Wilson (1951) were

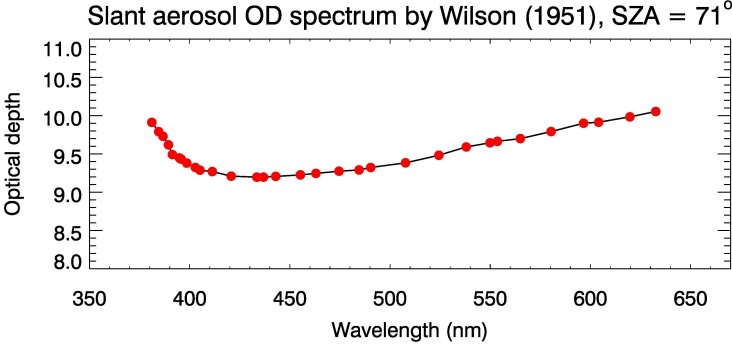

**Figure 7.** Spectrum of slant aerosol optical depth based on the analysis by Wilson (1951).

affected by the same or very similar systematic effects that largely cancelled out when taking the ratio of the two spectra. The results also demonstrate that a blue Sun can be reproduced even with a small relative spectral variation of the aerosol optical depth in the visible spectral range.

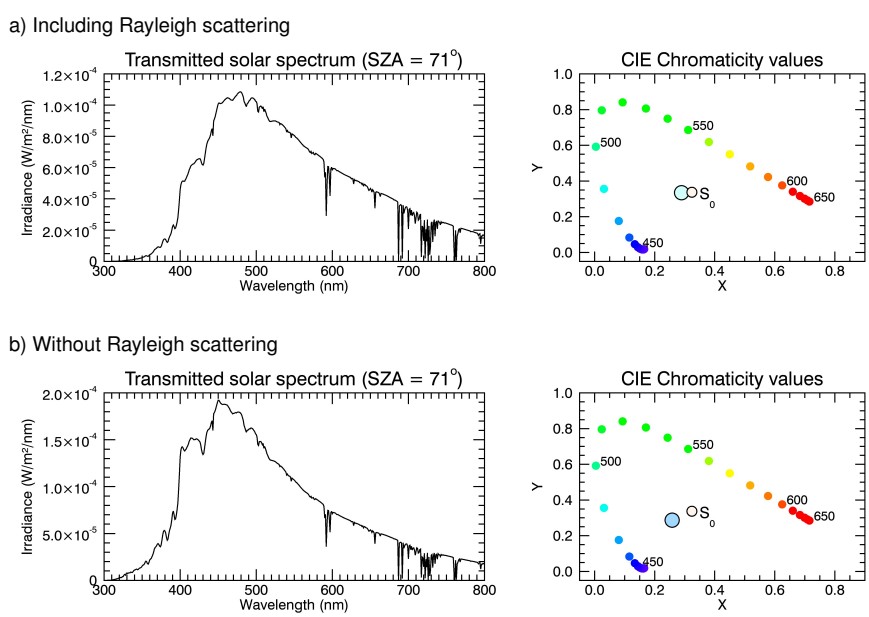

**Figure 8.** Similar to Figure 4, but for the aerosol optical depth and aerosol extinction spectrum taken from Wilson (1951) including Rayleigh scattering (top row) and without Rayleigh scattering (bottom row).

## 3.7 The role of absorption by $H_2O$

Several observers took measurements of the spectral distribution of transmitted solar radiation during the green or blue Sun events in the aftermath of the 1883 eruption of Krakatoa (Symons et al., 1888). The red part of the spectrum was found to be markedly suppressed. The Fraunhofer A line (at 760 nm, caused by $O_2$) could not be observed during the events and the Fraunhofer B line (at 687 nm, also caused by $O_2$) was difficult to observe and was not discernible for SZAs approaching 90° (Symons et al., 1888, p. 212). The observers reported unusually strong absorption by "water" (or water vapour), and absorption by water vapour was also discussed as a potential explanation of the blue Suns after the Krakatoa eruption (Symons et al., 1888).

The effect of $H_2O$ vapor absorption on the transmission spectra and the resulting colour of the Sun was also tested using SCIATRAN. For tropospheric $H_2O$ mixing ratios of 4% – which is often considered the upper limit in the troposphere – and an aerosol-free atmosphere, a blue colour of the Sun can not be reproduced (results not shown, but they are close to the results for the aerosol-free case shown in Fig. 3). Even for a tropospheric $H_2O$ mixing ratio of 40% – which is utterly unrealistic – the Sun appears yellowish and turns orange for large SZAs. Note that for the simulations the $H_2O$ mixing ratio was fixed at 4% and 40%, respectively, in the entire troposphere. Given the weak impact of $H_2O$ absorption on the simulated colour of the sun, we exclude absorption by $H_2O$ as a sole explanation for blue Suns. However, absorption of solar radiation by $H_2O$ may contribute to the blue colour, if scattering aerosols are present in the atmosphere. One may expect that high $H_2O$ mixing ratios lead to smaller aerosol optical depths required to produce a blue colour of the Sun, because $H_2O$ absorbs strongly in the red part of the visible spectrum.

## 3.8 The role of absorption by $O_3$

Since $O_3$ is an important absorber of solar radiation in the visible spectral range, it may potentially also affect the colour of the sun. Gedzelman and Vollmer (2009) qualitatively discussed the importance of absorption of radiation by $O_3$ for the occurrence of blue moons. We tested the effect of $O_3$ on the colour of the solar disk by varying the total $O_3$ column (TOC). Fig. 9 shows transmission spectra and chromaticity diagrams for a sample SZA of 80° and TOC values of 100 DU (top panel), 300 DU (middle panel) and 500 DU (bottom panel). The aerosol parameters are the same as in Fig. 4. The TOC values chosen cover essentially the entire range of possible values: 100 DU may occur under Antarctic $O_3$ hole conditions and 500 DU can be reached at high latitudes during Arctic winter. As follows from Fig. 9, absorption of solar radiation by $O_3$ in the Chappuis bands does not play a major role for the colour of the sun. In fact, the Chappuis bands are hardly visible in these plots. The bands are slightly visible for TOC = 500 DU if aerosol extinction is not considered (results not shown), but also in the aerosol-free case the effect of $O_3$ on the colour of the sun is small. The results do not change much if other SZAs are chosen. We also tested unrealistically large TOC values of up to 5000 DU (results not shown) and at these large values $O_3$ absorption has a discernible effect on the colour of the solar disk. Overall, we conclude that absorption of solar radiation in the Chappuis bands of $O_3$ is of minor importance for the colour of the sun, which is in partial contradiction to the qualitative considerations in Gedzelman and Vollmer (2009).

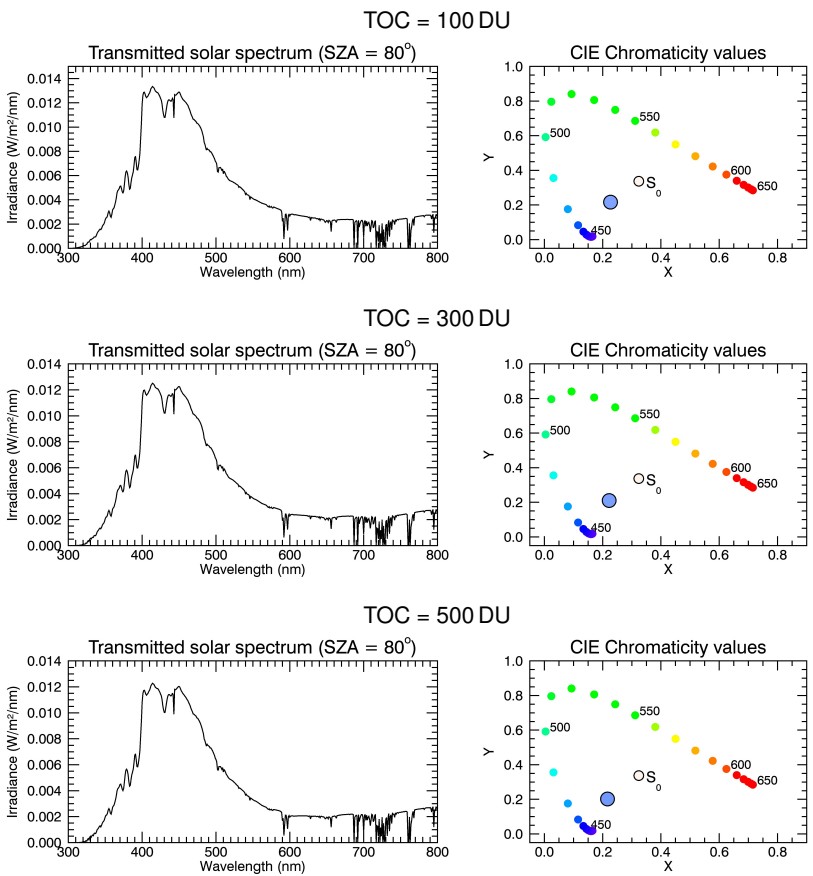

**Figure 9.** Similar to Fig. 4, but for different total ozone columns (top panel: 100 DU, middle panel: 300 DU, bottom panel: 500 DU).

## 3.9 Absorption by aerosols

In all previous attempts to explain the phenomenon of the blue Sun, absorption by aerosols has been dismissed and we believe
this was done for good reasons. Absorption coefficients of essentially all different types of aerosols, including mineral dust and
volcanic ash decrease with increasing wavelength in the visible spectral range. Bergstrom et al. (2007) report on measurements
of spectral absorption properties of the most important absorbing aerosol types, including biomass burning aerosols, desert dust,
pollution and mixtures, which consistently yield positive Ångström exponents, i.e., extinction coefficients (or optical depth)
decreasing with increasing wavelength. Patterson (1981) reported on measurements of the imaginary part of the refractive
index of volcanic ash emitted during the Mount St. Helens eruption in 1980. The imaginary part decreases with increasing
wavelength in the 300 nm – 700 nm spectral range, and so does the absorption cross section (or coefficient). Patterson et al.
(1983) reported similar measurements of the ash emitted by the El Chichon eruption in 1982 and found similar results.

Pollack et al. (1973) presented spectral variations of the "extinction coefficient" of different rock types and for some of the
rock types these coefficients increase with increasing wavelength in the visible spectral range. Please note that the presented

extinction coefficients are dimensionless and are defined in a different way (see also Eqn. 2 in Pollack et al. (1973)) than the quantity usually denoted extinction coefficient, having units of 1/km. In order to convert Pollack's extinction coefficient to the standard extinction coefficient one has to divide by the wavelength. If this is done then the extinction coefficients do not increase with wavelength any more.

Although absorption by aerosols is a rather unlikely explanation for the occurrence of blue sun or moon events, we performed SCIATRAN calculations considering absorption by aerosols. The calculations are based on the spectral dependence of the absorption aerosol optical depth of biomass burning aerosols measured during the SAFARI 2000 campaign as reported in Bergstrom et al. (2007) (see their Fig. 1). We assumed the Ångström exponent for the 325 – 1000 nm spectral range, i.e., $\alpha$ = 1.45. SCIATRAN simulations were then performed for different absorbing aerosol optical depths. Sample results for an aerosol absorption optical depth of 1.0 at 550 nm are shown in Fig. 10. Apparently, the effect of aerosol absorption is a further reddening of the solar disk – compared to the Rayleigh-only case shown in Fig. 3 – which is to be expected because the aerosol absorption optical depth decreases with increasing wavelength. This general behaviour is also observed for the other aerosol types discussed in Bergstrom et al. (2007).

## 4 Discussion

In this study we investigated the influence of aerosol particles with different size distributions and refractive indices on the colour of the Sun, as perceived by a ground-based human observer. Different factors were examined that may be able to evoke a shift in the transmitted solar spectral distribution towards a higher intensity at shorter wavelengths in the blue region of the visible spectrum. It is obvious that a suppression of red light — relative to the shorter wavelengths — is the key prerequisite for producing a blue Sun or a blue moon. In principle this may be caused by molecular extinction or extinction by aerosol particles.

The presented results suggest that absorption of solar radiation by $O_3$, $H_2O$ and aerosols are unlikely candidates to explain the occurrence of blue suns or moons. Our simulations confirm the basic notion of previous studies, that anomalous scattering in the visible spectral range caused by a suitably sized aerosol population is able to explain the occurrence of a blue or green Sun. A narrow aerosol particle size distribution with an appropriate mean or median radius will more easily lead to blue Suns compared to a broader size distribution. One general question is, how a narrow size distribution can be achieved in the real atmosphere. One possibility could be a size separation by sedimentation, perhaps accompanied by different wind directions at different altitudes, as already suggested by Gelbke (1951). The right combination of conditions for the occurrence of a blue Sun can be expected to arise only rarely, which provides an explanation for the infrequent observations of a blue Sun.

It should also be mentioned that SCIATRAN simulations in transmission mode treat only the directly transmitted solar radiation, while the contribution of scattering of solar photons into the field of view of the observer is not considered. In order to determine the relative contribution of scattered radiation, we ran SCIATRAN in scattering mode. It was found that scattered radiation is only a small fraction of the directly transmitted radiation for the cases considered. For example, for an aerosol optical depth of 1 (at 550 nm) and the aerosol parameters as in Fig. 4, the scattered radiation – from the solid angle

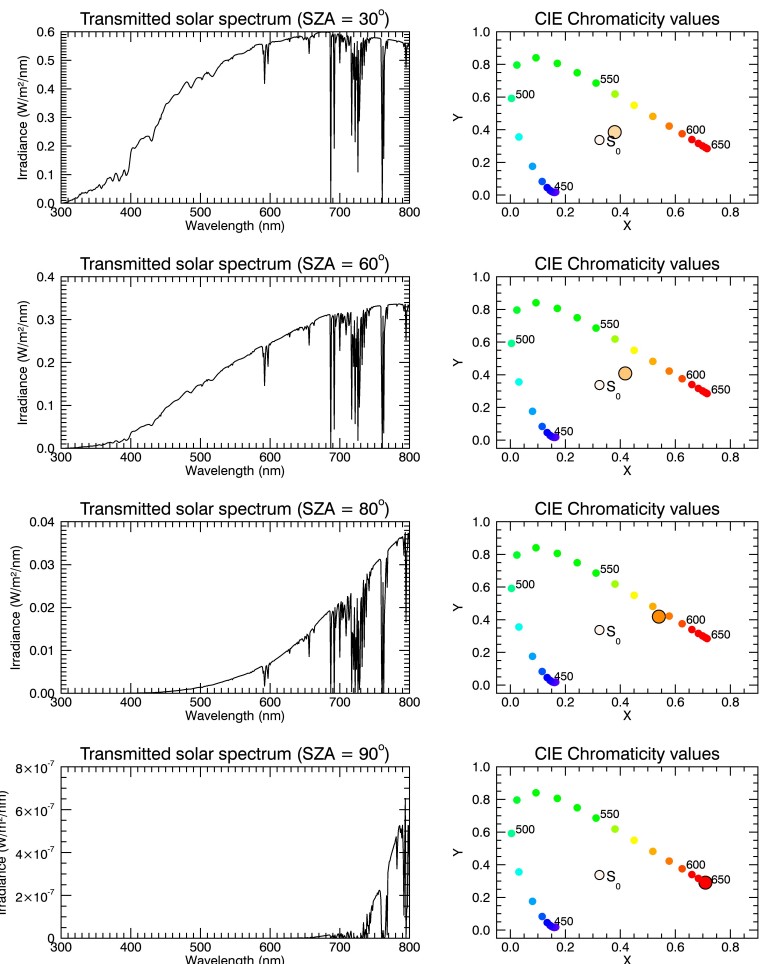

**Figure 10.** Similar to Fig. 4, but with aerosol absorption with an absorption aerosol optical depth of 1.0 at 550 nm and an Ångström exponent of $\alpha = 1.45$, as measured during the SAFARI 2000 campaign (Bergstrom et al., 2007).

corresponding to the solar disk – corresponds to about 0.04% of the transmitted radiation for a solar zenith angle of 30° and averaged over the visible spectral range. Without aerosols in the atmosphere this fraction decreases to about 0.0014%.

It is also important to point out, that colour perception is subject to multiple influencing factors. Modelling the colour of the solar disk as done here is a first important step but other factors especially in the context of the complex physiology of human colour perception such as light adaptation should not be neglected. For example, one not only needs to consider the colour of the Sun alone but also the surrounding sky colour. The contrast between those two different colours can influence the colour perception of the Sun itself. The phenomenon has been referred to as simultaneous contrast (see, e.g., chapter 11

in Brainard and Stockman (2010)). The surroundings of a coloured area induce an apparent colour change to the surrounding area's complement colour. For the Sun on a blue sky this would induce a slightly more yellow tinge to the solar disk, since

yellow is the complement colour to blue (Choudhury, 2014). This would imply a reduction in the phenomenon's perception. Furthermore, the brightness or intensity of the light can influence the perception of the colour itself, as the different cones and rods in the human eye have different light adaptations and different sensitivities in changing light conditions (Simon, 2008).

This is probably negligible when the Sun's brightness is much greater than the surrounding sky. Then the sky can appear darker due to the overwhelming brightness of the sun, but the Sun's colour is most likely less effected. In a more opaque atmosphere where the disk of the Sun is clearly visible, simultaneous contrast should be more carefully considered. Due to rarity of the phenomenon and the fact that only few photographs of blue suns are available, the surrounding sky's colour is hard to consider. Simulating the whole sky's colour can be a next step towards a complete picture of the phenomenon.

In this context it is also worth mentioning that Gelbke (1951) discussed a potential influence of contrast-induced blueing of the solar disk – possibly caused by the fact that the blue sun photographed in 1950 from Greifswald was surrounded by reddish-brownish altocumulus clouds. However, blue suns were also observed without being surrounded by the complementary colours of blue. In addition, blue moons were observed during the night, i.e. without any additional colouring of the sky. Still, contrast effects may certainly affect the apparent colour of the Sun.

The results presented so far were mostly for an aerosol layer in $6 - 8\,\mathrm{km}$ altitude, which is certainly arbitrary. Simulations were also carried out for aerosol layers in $2 - 4\,\mathrm{km}$ and $10 - 12\,\mathrm{km}$ altitude. We generally found little dependence of the results on layer altitude, except for SZAs approaching $90°$. At these large SZAs the blueing effect on the Sun gets more pronounced the lower the aerosol layer is (results not shown). It is beyond the scope of this study to present all cases in detail.

## 5 Conclusions

We investigated possible reasons for the occurrence of a blue Sun, including anomalous scattering by suitably sized aerosols, aerosol absorption and absorption by water vapour and $O_3$. This study included – for the first time to our best knowledge – a comprehensive treatment of Rayleigh scattering in simulations of a blue Sun. Depending on the specific aerosol and observational parameters (SZA), Rayleigh scattering was found to have a significant impact on the simulated colour of the Sun. Anomalous extinction is more pronounced for narrow aerosol particle size distributions (given the right aerosol size), but

the investigations based on the aerosol optical depth retrieved by Wilson (1951) demonstrated that a relatively weak increase in aerosol extinction with wavelength in the visible spectral range also allows simulating a blue Sun. This implies that blue Suns can also be produced for somewhat broader size distributions. Our results indicate that an aerosol optical depth of about $0.5$ is sufficient to explain a blue Sun, if the aerosol size parameters are chosen in an optimal way. If Rayleigh scattering is neglected – which is unrealistic, but has been done in many earlier studies – an aerosol optical depth of only $0.1$ is sufficient to yield a

blue Sun. Absorption by water vapour – also proposed as a potential explanation for the blue Sun phenomenon – was shown to be a very unlikely explanation. Absorption of radiation by $O_3$ was also found to have only a minor influence on the colour of the sun. This study confirms the result of several other studies, that anomalous extinction by aerosol particles of a specific size and size distribution allows explaining the occurrence of blue or green Suns. However, Rayleigh scattering should not be neglected.

*Author contributions.* CvS outlined the project, NW carried out the initial SCIATRAN simulations with guidance by AR and implemented the colour display approach. NW also wrote an initial version of the manuscript. AL and CvS improved the analyses and extended the manuscript. All authors discussed, edited and proofread the paper.

*Competing interests.* The authors declare that they have no competing interests.

*Code availability.* The SCIATRAN radiative transfer model can be downloaded from the following website: https://www.iup.uni-bremen.de/sciatran/.

*Acknowledgements.* This work was supported by the Deutsche Forschungsgemeinschaft (project VolARC of the DFG research unit VolImpact FOR 2820, grant no. 398006378). We are indebted to the Institute of Environmental Physics of the University of Bremen – particularly to Dr. Vladimir Rozanov and Prof. Dr. John P. Burrows FRS – for access to the SCIATRAN radiative transfer model.

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
