# Peer review of "On the phenomenon of the blue Sun"

_Climate of the Past, 2020_

## Referee Comment (RC1) · Anonymous Referee #1 · 28 Oct 2020

**1 General Comments**

This manuscript examines the unusual atmospheric conditions under which a blue sun can occur. The analysis is similar to previous work, in particular that by Horvath et al., (1993), but the authors extend the analysis by including more complete radiative transfer simulations. Results generally agree with previous work, but the authors find that higher aerosol optical depths are required when a realistic atmosphere is accounted for. Overall, the manuscript presents a well written description of the topic and analysis performed. I would recommend publication after minor corrections.

**2 Specific Comments**

Line 126: Why is a reference of 350nm chosen? This seems generally beyond the limits of human vision. If "maximum anomalous extinction" is the ratio of red/blue extinction this seems an odd choice given the sensitivity curves in Figure 1.

Line 130: Along those lines, please define "maximum anomalous extinction".

Line 149: A brief description of what "int - transmission - CDI" means would be useful.

Line 149: Ehlers (2014) found that the forward scattering by aerosols was an important contribution to the blue sun on Mars. Do the radiative transfer calculations take this into account and is this not an important factor for the cases investigated here?

Line 223 – 227: Is a particle size assumed in this analysis, or is the wavelengthdependent extinction used directly?

Figure 7: How does this compare to the wavelength dependence of the particle sizes used in the previous analysis?

Line 237: "... despite the questionable spectral signatures in the solar transmission spectra." Although the included figures make it clear, I think this wording is a bit ambiguous as to whether your simulated spectra reproduce this feature. Just a suggestion, but I would rephrase to something like "... although the minimum in the solar transmission spectra near 490nm was not reproducible"

Line 250: Do the water vapour simulations fix the entire mixing ratio profile at 4/40

Line 276: It is not clear to me from Pollack (1973) that the absorption could not be increasing for certain particle makeup. While clearly not necessary for a blue sun, I don't think this work has shown that it is an unlikely contributor, especially given the unknown makeup and rarity of the events.
**3** Technical Corrections**

Figure 1: should the x, y, z labels have bars over them?

Line 191: remove "of" from "aerosol with of a sufficiently small..."

Line 223: "...variation is with about 10

---

## Referee Comment (RC2) · Anonymous Referee #2 · 16 Dec 2020

Report on: On the phenomenon of the blue Sun
Author(s): Nellie Wullenweber et al.
submitted to Climate of the Past (CP), https://doi.org/10.5194/cp-2020-117

Overall I expect that the authors will adequately deal with my comments, therefore I assume that the paper will be acceptable after a revision.

So my official recommendation is: some amendments needed

The paper presents an extensive theoretical study of how aerosol and Rayleigh scattering can modify the perceived color of the sun. It does so using aerosol layers of given optical thickness containing aerosol size distributions with variable mean size and width, as defined by log normal distributions. As expected, the results indicate that anomalous scattering ($\sigma$ increases with $\lambda$, which can give rise to blue suns or moons) are rather rare phenomena and require special conditions such as sometimes happening after volcanic eruptions or intense forest fires. The authors nicely describe the effects of optical depths including Rayleigh scattering which was previously mostly neglected when explaining blue suns.

Overall the paper is well structured and serves its purpose. I have however several suggestions and questions which should be dealt with.

First general comment: Starting to read I immediately missed a reasoning why absorption by aerosols is neglected. I had to wait for Sect. 4 where – in a more or less vague and qualitative manor – it was discussed that absorption has been dismissed due to results of several studies. I cannot help but think that instead of qualitatively arguing, you should compute maybe just one example for typical imaginary parts from the literature and compare the results to the absorption free case. Then any reader would accept your arguments. Right now I have just to believe them which is unsatisfactory.

Second general comment: Color perception is more than just the xy-coordinate in a chromaticity diagram. It is very helpful to plot the color, however, in real observations the influence of attenuation is dramatic when moving for 30° SZA to 90°SZA and one must consider the influence of changing contrast. The least I would expect is to not only show the color of the sun, but also how it´s relative brightness changes. Even better would be to add a short discussion of how color perception is expected to be influenced by the variation in brightness and contrast.

Third general comment: I missed a discussion concerning the influence of ozone absorption (Chappuis bands) on the color, in particular for large SZA. It had been shown e.g. for lunar eclipses that blue color during totality can be influenced by ozone (Appl. Opt. 47, No. 34 / H149 (2008)). The effect of ozone has also been briefly discussed for blue moons (see literature). I assume that your model does also allow to assume an ozone distribution and check how / if results change.

In the following I mention some additional thoughts which I had when reading the text.

line 77, see first general comment

line 91 Sect. 2.2: of course this depends on the physics/optics knowledge of the general reader of the journal, but to my opinion, the topic of color as treated in the section is basic textbook

knowledge. This section can be shortened appreciably including deleting Fig. 1. I suggest to just quote respective textbooks. Reason: those who are not familiar with chromaticity diagrams and respective definitions will not get a better knowledge when reading this condensed textbook knowledge and those familiar with the topic do not need it. I would just keep the last sentences, i.e. lines 110-117

line 128; discussion of figure 2: Maybe I am old fashioned here, but I personally think looking at Fig. 2 alone makes it harder to see the point. I suggest that you should add the classic diagram of extinction efficiency versus size parameter which explains anomalous extinction at one glance. Having this in mind greatly helps to understand your admittedly nice representation of the same content in Fig. 2.

Line 147: you mention standard atmospheric trace gases are used, is ozone included ? see third general comment

Line 161: have you used the exponent 4.00 for Rayleigh scattering or averaged 4.08 which includes dispersion effects of air (e.g. Young, Phys Today Jan 1982, p2-8)?

Line 72: question: is the vertical optical depth of 1 only the aerosol or the total optical depth ? please clarify!

Line 152: Starting Sect. 3.2, discussion of diagrams 3-6 and 8: see second general comment.

Line 220: from fig. 7 I estimated a change in attenuation from 400 nm to 630 nm from around $10^{-4}$ to $4.3 \cdot 10^{-5}$ i.e. a factor of 2.3 between blue and red. You describe the change in depth as only being 10%, that is true, but misleading. It is the factor of 2.3 difference for the radiometric quantities entering the eye of an observer which is relevant. Maybe you could amplify the change by plotting from 8 to 11 rather than 6 to 12.
And of course, the attenuation of more than 10,000 also means that sun is not very bright, though still bright enough for color perception. One may compare this to totality of a solar eclipse where attenuation with regard to daylight is around $10^{-5}$.

Line 222: "were" only available (you refer to Wilsons data)

Line 223: maybe better to write … variation with about 10% change is not …

Line 265: see general comment 1

**references:**
I propose that the authors carefully check all refs. I found e.g. one misspelling.

Line 325. Should probably be Dietze (e missing), as G. Dietze in his book on atmospheric optics from 1957 mentions blue suns.

I had expected some other general standard textbook refs. which at least shortly discuss blue suns in the context of Mie scattering such as Van de Hulst, Light scattering by small particles from 1957 or Bohren/Huffman, Absorption and scattering of light by small particles, Wiley 1983.

In addition, Gedzelman / Vollmer Twice in a blue moon, Weatherwise Sept/Oct 2009, 28-35, discussed blue moons including the role of ozone and also a but of optical depth discussion

---

## Author Comment (AC1) · 9 Feb 2021

Reply to comments by anonymous referee #1

General Comments

This manuscript examines the unusual atmospheric conditions under which a blue sun can occur. The analysis is similar to previous work, in particular that by Horvath et al., (1993), but the authors extend the analysis by including more complete radiative transfer simulations. Results generally agree with previous work, but the authors find that higher aerosol optical depths are required when a realistic atmosphere is accounted for. Overall, the manuscript presents a well written description of the topic and analysis performed. I would recommend publication after minor corrections.

**Reply: We thank the reviewer for his/her encouraging comments. We imple-**

[Figure]

**mented essentially all the changes requested by the reviewer (see detailed responses below).**

Specific Comments

Line 126: Why is a reference of 350nm chosen? This seems generally beyond the limits of human vision. If "maximum anomalous extinction" is the ratio of red/blue extinction this seems an odd choice given the sensitivity curves in Figure 1.

**Reply: This choice was somewhat arbitrary and the reviewer is correct that it was probably not the best choice. Although the main message of the Figures is essentially the same, we changed the reference wavelength to 400 nm.**

Line 130: Along those lines, please define "maximum anomalous extinction".

**Reply: We replaced this sentence by the following:**

**"As can be seen in Fig. 2, the strongest increase in extinction coefficient with increasing wavelength in the visible spectral range (or maximum anomalous extinction) is obtained under the assumptions made for median radii in the 400 – 700 nm range, with the specific values depending on the assumed refractive index."**

**We hope that this change clarifies the meaning.**

Line 149: A brief description of what "int - transmission - CDI" means would be useful.

**Reply: Good idea! We adapted the text and added more information here.**

Line 149: Ehlers (2014) found that the forward scattering by aerosols was an important contribution to the blue sun on Mars. Do the radiative transfer calculations take this into account and is this not an important factor for the cases investigated here?

**Reply: Thanks for this comment. Ehlers et al. (2014) found that scattering was important for the blue glow around the solar disk on Mars, but they also stated**

that the slightly blue colour of the solar disk was caused by wavelength selective extinction (that's the effect our paper deals with; Ehlers et al. call this "bluing"). These effects have to be distinguished. Ehlers et al. did not state that scattering by aerosols was the cause of the blue colour of the solar disk.

**For the SCIATRAN simulations presented in our paper only the transmission is calculated, scattering is not considered. In the revised version of the paper we will include a comparison of the contributions of transmission and scattering to the total intensity.**

Line 223 – 227: Is a particle size assumed in this analysis, or is the wavelength-dependent extinction used directly?

**Reply: The optical depth spectrum published by Wilson was directly used as input for the SCIATRAN simulations. We adapted the text to make this point clearer.**

Figure 7: How does this compare to the wavelength dependence of the particle sizes used in the previous analysis?

**Reply: The wavelength dependence in Fig. 7 is significantly weaker than for the results shown, e.g. in Fig. 4, where close to the optimum particle size parameters were chosen to produce a blue sun (see also Fig. 2 of the online version or Fig. 1 in the revised version of the manuscript). We added some text to the paper to mention this difference.**

Line 237: "...despite the questionable spectral signatures in the solar transmission spectra." Although the included figures make it clear, I think this wording is a bit ambiguous as to whether your simulated spectra reproduce this feature. Just a suggestion, but I would rephrase to something like "...although the minimum in the solar transmission spectra near 490nm was not reproducible"

**Reply: OK, text changed as suggested.**

Line 250: Do the water vapour simulations fix the entire mixing ratio profile at 4/40

**Reply: Yes, the water vapour mixing ratios in the troposphere were 4 / 40% at all altitudes. We added this piece of information to the manuscript.**

Line 276: It is not clear to me from Pollack (1973) that the absorption could not be increasing for certain particle makeup. While clearly not necessary for a blue sun, I don't think this work has shown that it is an unlikely contributor, especially given the unknown makeup and rarity of the events

**Reply: Thanks for this comment. We read Pollack (1973) again carefully and realized that our explanation in the paper is not entirely correct. Our main conclusion is, however, not affected. Fig. 1 of Pollack shows the spectral dependence of the "Extinction coefficient" and this quantity is increasing with increasing wavelength in the visible spectral range. However, this extinction coefficient is dimensionless and is different from the standard definition of the extinction coefficient, having a dimension of 1/length. Please have a look at equation 2 in Pollack, which shows Beer-Lambert's law. Here the wavelength appears in the denominator of the exponent making the extinction coefficient indeed dimensionless. In order to convert Pollack's extinction coefficient to the standard extinction coefficient one has to divide by the wavelength. If this is done then the extinction coefficients don't increase with wavelength any more.**

**Our explanations in lines 275 – 279 of the online manuscript wrongly discuss the spectral dependence of the refractive index, not the spectral dependence of Pollack's "extinction coefficient", we apologize. This is now corrected.**

**Regarding aerosol absorption as a possible source of blue suns we also carried out simulations including aerosol absorption (as suggested by reviewer 2) and added a section to the manuscript.**

Technical Corrections

Figure 1: should the x, y, z labels have bars over them?

**Reply: The reviewer is correct; the bars were missing. Following the recommendation by reviewer 2, we deleted this Figure, however.**

Line 191: remove "of" from "aerosol with of a sufficiently small..."

**Reply: Thank you – corrected.**

Line 223: "...variation is with about 1

**Reply: OK, changed!**

---

## Author Comment (AC2) · 9 Feb 2021

Reply to comments by referee #1

Referee comments:

Overall I expect that the authors will adequately deal with my comments, therefore I assume that the paper will be acceptable after a revision.

So my official recommendation is: some amendments needed

The paper presents an extensive theoretical study of how aerosol and Rayleigh scattering can modify the perceived color of the sun. It does so using aerosol layers of given optical thickness containing aerosol size distributions with variable mean size and width, as defined by log normal distributions. As expected, the results indicate that anomalous scattering ($\sigma$ increases with $\lambda$, which can give rise to blue suns or

moons) are rather rare phenomena and require special conditions such as sometimes happening after volcanic eruptions or intense forest fires. The authors nicely describe the effects of optical depths including Rayleigh scattering which was previously mostly neglected when explaining blue suns. Overall the paper is well structured and serves its purpose. I have however several suggestions and questions which should be dealt with.

**Reply: We thank the reviewer for his/her very constructive comments. As described in more detail below, we essentially followed all the suggestions by the reviewer.**

First general comment: Starting to read I immediately missed a reasoning why absorption by aerosols is neglected. I had to wait for Sect. 4 where – in a more or less vague and qualitative manor – it was discussed that absorption has been dismissed due to results of several studies. I cannot help but think that instead of qualitatively arguing, you should compute maybe just one example for typical imaginary parts from the literature and compare the results to the absorption free case. Then any reader would accept your arguments. Right now I have just to believe them which is unsatisfactory.

**Reply: This is a very good idea and we now included a new subsection on the importance of absorption by aerosols and discussed one example with a typical absorption signature of aerosols.**

Second general comment: Color perception is more than just the xy-coordinate in a chromaticity diagram. It is very helpful to plot the color, however, in real observations the influence of attenuation is dramatic when moving for 30° SZA to 90°SZA and one must consider the influence of changing contrast. The least I would expect is to not only show the color of the sun, but also how it's relative brightness changes. Even better would be to add a short discussion of how color perception is expected to be influenced by the variation in brightness and contrast.

**Reply: This is a good point, too, and in the submitted version of the manuscript**

**we didn't really discuss this aspect. The irradiance variations are actually shown in some of the Figures (3, 4, 6, 8 of the discussion paper), but we didn't discuss them. We now added paragraphs on possible effects of brightness changes and on potential contrast effects to the discussion section.**

Third general comment: I missed a discussion concerning the influence of ozone absorption (Chappuis bands) on the color, in particular for large SZA. It had been shown e.g. for lunar eclipses that blue color during totality can be influenced by ozone (Appl. Opt. 47, No. 34 / H149 (2008)). The effect of ozone has also been briefly discussed for blue moons (see literature). I assume that your model does also allow to assume an ozone distribution and check how / if results change.

**Reply: Thank you for this suggestion. Yes, we can easily test the effect of ozone (ozone is considered in the results in the submitted version of the paper) and we have performed a series of new simulations with varying amounts of ozone.**

**If an aerosol layer with an OD of 1 is considered (as in Fig. 4 of the online manuscript) the effect of changing the total ozone column between 100 DU (unrealistically low except for ozone hole conditions) and 500 DU is very minor. The transmission spectra are certainly affected – also in an aerosol-free case – but the overall shape of the transmission spectra is not changed to the extent to cause a significant change in the colour of the sun. We also tested unrealistically large ozone columns (1000, 2000 and 5000 DU) that lead to large differences in the transmission spectra, and also the colour of the sun. For Ozone columns up to 500 DU differences are difficult to identify visually.**

**We added a new subsection (3.8) to the manuscript discussing the effect of ozone absorption on the transmission spectra and the resulting colours of the sun. A few sample simulations results will be included, too.**

In the following I mention some additional thoughts which I had when reading the text.

line 77, see first general comment

**Reply: As mentioned in our reply to the first general comment above, we followed the reviewer's suggestion and simulated the effect of aerosol absorption on the resulting colors of the solar disk. Line 77 was adjusted correspondingly.**

line 91 Sect. 2.2: of course this depends on the physics/optics knowledge of the general reader of the journal, but to my opinion, the topic of color as treated in the section is basic textbook knowledge. This section can be shortened appreciably including deleting Fig. 1. I suggest to just quote respective textbooks. Reason: those who are not familiar with chromaticity diagrams and respective definitions will not get a better knowledge when reading this condensed textbook knowledge and those familiar with the topic do not need it. I would just keep the last sentences, i.e. lines 110-117

**Reply: We followed the reviewer's suggestion and reduced this section significantly and referred to a standard text book and some other sources. We did not delete the lines before 110 entirely, because it is our experience that many colleagues have problems interpreting the chromaticity diagram without additional information.**

line 128; discussion of figure 2: Maybe I am old fashioned here, but I personally think looking at Fig. 2 alone makes it harder to see the point. I suggest that you should add the classic diagram of extinction efficiency versus size parameter which explains anomalous extinction at one glance. Having this in mind greatly helps to understand your admittedly nice representation of the same content in Fig. 2.

**Reply: OK, we added an additional plot as suggested by the reviewer.**

Line 147: you mention standard atmospheric trace gases are used, is ozone included ? see third general comment

**Reply: Yes, ozone is included and we now mention the considered trace gases explicitly in the manuscript.**

Line 161: have you used the exponent 4.00 for Rayleigh scattering or averaged 4.08 which includes dispersion effects of air (e.g. Young, Phys Today Jan 1982, p2-8)?

**Reply: We used the Bates (1984) formula of the Rayleigh scattering cross section. We determined the spectral exponent by fitting a power law to the spectral dependence of the Rayleigh extinction cross section in the 400 nm – 700 nm and obtained the value: 4.0836 $\pm$ 0.0067. We now mention explicitly in the paper that the Bates formula is used.**

**Reference:**

**D.R. Bates, Rayleigh scattering by air, Planetary and Space Science, Volume 32, Issue 6, Pages 785-790, https://doi.org/10.1016/0032-0633(84)90102-8, 1984.**

Line 172: question: is the vertical optical depth of 1 only the aerosol or the total optical depth ? please clarify!

**Reply: The vertical optical depth of 1 refers to the aerosols only. We replaced "optical depth" by " aerosol optical depth" to make this clear.**

Line 152: Starting Sect. 3.2, discussion of diagrams 3-6 and 8: see second general comment.

**Reply: We added a discussion of brightness and contrast effects to the manuscript.**

Line 220: from fig. 7 I estimated a change in attenuation from 400 nm to 630 nm from around 10-4 to 4.3 10-5 i.e. a factor of 2.3 between blue and red. You describe the change in depth as only being 10%, that is true, but misleading. It is the factor of 2.3 difference for the radiometric quantities entering the eye of an observer which is relevant. Maybe you could amplify the change by plotting from 8 to 11 rather than 6 to 12. And of course, the attenuation of more than 10,000 also means that sun is not very bright, though still bright enough for color perception. One may compare this to totality of a solar eclipse where attenuation with regard to daylight is around 10-5.

**Reply: You are certainly correct that the change in radiance is relevant, not the change in OD. We adjusted the statement and plotted the Figure again with the y-range suggested by the reviewer.**

Line 222: "were" only available (you refer to Wilsons data)

**Reply: changed.**

Line 223: maybe better to write . . . variation with about 10% change is not . . .

**Reply: this statement was already corrected following the comment by reviewer 1.**

Line 265: see general comment 1

**Reply: We included a new subsection on the effect of absorption by aerosols and included sample results of transmission simulations with aerosol absorption, as suggested by the reviewer.**

references:

I propose that the authors carefully check all refs. I found e.g. one misspelling.

**Reply: We proofread the references several times and corrected a few minor errors.**

Line 325. Should probably be Dietze (e missing), as G. Dietze in his book on atmospheric optics from 1957 mentions blue suns.

**Reply: Thank you for catching this, the correct spelling is "Dietze".**

I had expected some other general standard textbook refs. which at least shortly discuss blue suns in the context of Mie scattering such as Van de Hulst, Light scattering by small particles from 1957 or Bohren/Huffman, Absorption and scattering of light by small particles, Wiley 1983.

**Reply: Okay, we now also cite Van de Hulst and Bohren & Huffman.**

In addition, Gedzelman / Vollmer Twice in a blue moon, Weatherwise Sept/Oct 2009, 28-35, discussed blue moons including the role of ozone and also a but of optical depth discussion.

**Reply: Thanks for this suggestion, we were not aware of this article. It is now also cited in our manuscript.**

——————————————————